# Enhancement of the Anti-Stokes Fluorescence of Hollow Spherical Carbon Nitride Nanostructures by High Intensity Green Laser

**DOI:** 10.3390/nano11102529

**Published:** 2021-09-28

**Authors:** Pavel V. Zinin, Tayro E. Acosta-Maeda, Anupam K. Misra, Shiv K. Sharma

**Affiliations:** 1Scientific Technological Center of Instrumentation, Russian Academy of Sciences, Butlerova Str. 15, 117342 Moscow, Russia; 2Hawaii Institute of Geophysics and Planetology, University of Hawaiʻi, 1680 East-West Road, Honolulu, HI 96822, USA; tayro@hawaii.edu (T.E.A.-M.); anupam@hawaii.edu (A.K.M.); sksharma@soest.hawaii.edu (S.K.S.)

**Keywords:** anti-Stokes, *g*-C_3_N_4_, *s*-C_3_N_4_, fluorescence

## Abstract

Fluorescence spectra of graphitic (*g*-C_3_N_4_) and spherical (*s*-C_3_N_4_) modifications of carbon nitride were measured as a function of green pulsed (6 ns-pulse) laser intensity. It was found that the intensity of the laser increases the maximum of the fluorescence shifts towards the anti-Stokes side of the fluorescence for *s*-C_3_N_4_ spherical nanoparticles. This phenomenon was not observed for *g*-C_3_N_4_ particles. The maximum of the anti-Stokes fluorescence in *s*-C_3_N_4_ nanoparticles was observed at 480 nm. The ratio of the intensity of the anti-Stokes peak (centered at 480 nm) to that of the Stokes peak (centered at 582 nm) was measured to be I_484/582_ = 6.4 × 10^−3^ at a low level of intensity (5 mW) of a green pulsed laser, whereas it rose to I_484/582_ = 2.27 with a high level of laser intensity (1500 mW).

## 1. Introduction

Recently, graphite-like carbon nitrides have begun to attract attention because of their possible use in tasks such as splitting water by optical methods [1], as metal-free catalysts [2], and for the degradation of organic pollutants in water under the influence of visible light [3]. It was also found that that graphitic modifications of C_3_N_4_ are extremely fluorescent under laser irradiation [4,5]. The high yield of fluorescence, small size, intrinsic optical properties, low toxicity, and useful non-covalent interactions make graphitic carbon nitride nanoparticles an effective optical sensor to detect various ions [6]. They also can be used as a traceable and pH-responsive drug delivery system [7]. 

Despite the increasing number of publications devoted to the use of the optical properties of graphite-like carbon nitrides in various technological applications, a systematic study of the influence of the nanostructure on optical properties of C–N compounds has not been conducted. We know of only a few studies that investigated the fluorescence of different modifications of *g*-C_3_N_4_ [8,9]. In Ref. [8], it was found that the intensity of the fluorescence excited in *s*-C_3_N_4_ is more than three orders higher than that for *g*-C_3_N_4_ modification. Study of the fluorescence of *s*-C_3_N_4_ nanoparticles also demonstrated that fluorescence of *s*-C_3_N_4_ has a high intensity anti-Stokes part. Anti-Stokes fluorescence has many applications including nanoscale thermometry [10], laser cooling [11,12], and anti-Stokes fluorescence imaging of microscale thermal fields in thin films [13]. However, the non-linear behavior of the anti-Stokes fluorescence in graphitic carbon nitride has not been investigated. This paper looks at the non-linear effect of the pulsed laser excitation with different graphitic C_3_N_4_ structures including *s*-C_3_N_4_ and disordered *g*-C_3_N_4_ modifications on anti-Stokes fluorescence. Single-layered *g*-C_3_N_4_ quantum dots have previously been used for two-photon fluorescence imaging of cellular nucleus [14,15]. 

## 2. Materials and Methods

Both modifications of C_3_N_4_ were prepared by a high-temperature polycondensation reaction using lithium nitride (Li_3_N) as a nitridation and cross-linking agent and cyanuric chloride as an s-triazine building block [16,17,18]:
C_3_N_3_Cl_3_ + Li_3_N → C_3_N_4_ + 3LiCl.(1)


Carbon nitride hollow spheres, with diameters ranging from 20 microns to as few as 30 nanometers were added to a reactor as a substrate with nano-size silica spheres [18]. The reported electron microscopy data on the hollow spheres “suggest their multiwalled nanostructure, built by disorderly stacked C_3_N_4_ curved layers are assembled from triazine rings and nitrogen bridges of pyramidal structure” [18]. Spherical nanoparticles are aggregated to submicron clusters with an average size of the particle of 527 ± 80 nm determined by the dynamic light scattering measurements [8]. Disordered graphitic C_3_N_4_ (*g*-C_3_N_4_) was obtained under the same conditions [16] as for the synthesis of *s*-C_3_N_4_, however, nano-size silica spheres were not used as substrate. Results of the detailed characterization of *g*-C_3_N_4_ can be found elsewhere [16].

All spectra were acquired with either 200 ns or 500 ns gate widths over one laser pulse excited by 532 nm Nd-YAG laser (Lumibird, Bozeman, MT, USA), 15 Hz, 6 ns-pulse. The laser spot was approximately 5 mm in diameter. A detailed description of the system can be found elsewhere [19].

## 3. Results

The spectra of the fluorescence for *s*-C_3_N_4_ powders as function of laser power are shown in Figure 1a. The increase in laser intensity leads to a gradual decrease in Stokes intensity of the fluorescence of the *s*-C_3_N_4_ and an increase in the intensity of the anti-Stokes fluorescence. The maximum of the anti-Stokes fluorescence in *s*-C_3_N_4_ nanoparticles was observed at 480 nm. The anti-Stokes component had a minimum threshold of ~100 mW for this experimental setup. The Stokes (regular) component increases up to laser intensity 98 mW and then starts decreasing. The ratio of the intensity of the anti-Stokes peak (centered at 480 nm) to that of the Stokes peak (centered at 582 nm) was measured to be I_484/582_ = 6.4 × 10^−3^ at a low level of intensity (5 mW) of a green pulsed laser, whereas it rises to I_484/582_ = 2.27 with a high level of laser intensity (1500 mW). The powder is bright blue, which indicates that the intensity of the fluorescence has moved to the lower wavelength region under high power illumination. 

Because there is such a large difference in the intensity of fluorescence between the *s*-C_3_N_4_ and *g*-C_3_N_4_ forms of carbon nitride [8], we compared the fluorescence of these forms under pulsed laser illumination. The results of the measurements of the *g*-C_3_N_4_ are shown in Figure 2. 

## 4. Discussion

High intensity of anti-Stokes radiation was detected in *s*-C_3_N_4_ [8] but not in the quantum dots of carbon nitride [5]. The high intensity of the anti-Stokes radiation and anomalous fluorescence are associated with the spherical shape of the *s*-C_3_N_4_ nanoparticles. The non-linear properties are also a characteristic feature of *s*-C_3_N_4_ nanoparticles (Figure 1a and Figure 2). Currently, a physical model that explains the difference of the optical properties of the *s*-C_3_N_4_ and *g*-C_3_N_4_ materials in not available. However, here, we present experimental data that can shed light on the origin of the anomalous behavior of *s*-C_3_N_4_ nanomaterials. 

Figure 3 shows the behavior of the fluorescence of the *s*-C_3_N_4_ nanoparticles excited by CW lasers with different wavelengths. It demonstrates that there is only one center of the emission peak at 480 nm when illumination of the *s*-C_3_N_4_ was conducted with a 266 nm laser (Figure 3b). The position of the fluorescence peak with a green laser (514 nm) is around 582 nm, which is similar to that in Ref. [8]. 

Two emission peaks were found at 405 nm and 480 nm [14]. The emission band centered at 480 nm is attributed to the transition between the lone-pair (LP) valence band and the π* conduction band [20,21]. We attribute the emission center in *s*-C_3_N_4_ nanoparticles to the transition between π and the π* conduction band [22]. 

Two-photon fluorescence has already been detected in *g*-C_3_N_4_ single-layered quantum dots [14]. This system has a large two-photon absorption cross section. We believe that the nature of the non-linear effect in *s*-C_3_N_4_ nanoparticles is different. 

Figure 4 shows the log–log curve of the fluorescence of *s*-C_3_N_4_ nanoparticles. The slope is close to one, whereas for two photon fluorescence of the *g*-C_3_N_4_ single-layered QDs it is close to two [14]. This indicates that the mechanism of the enhancement of the anti-Stokes radiation is different from that of *g*-C_3_N_4_ single-layered QDs and might be related to the upconversion processes [23]. It has long been recognized that optical pumping of a medium accompanied by spontaneous anti-Stokes scattering can lead to the lowering of its temperature [11,24]. Two photon upconversion (UC) processes involve the sequential absorption of two or more photons. Thus, UC processes are different from the multiphoton process where the absorption of photons occurs simultaneously [25]. 

The log–log plot in Figure 4 indicates that in the case of *s*-C_3_N_4_ nanoparticles we are dealing with the phonon-assisted anti-Stokes excitation process discussed in [10]. According to [10], a photon at the long-wavelength tail of the absorption spectrum excites an electron from a thermally populated first vibronic state of the electronic ground state to the bottom manifold of an excited electronic state. The system can then return to the ground state via spontaneous emission of an upconverted photon. This photon-assisted anti-Stokes excitation process scales exponentially with temperature [10]. The tentative schematic energy level diagram of the PL emission from *s*-C_3_N_4_ is presented in Figure 5.

This diagram can be considered as a combination of the simplified energy level diagram described in Ref. [22] and that discussed in Ref. [10]. When the laser intensity is low, we observe stock fluorescence excited by a green laser with a peak centered at 582 nm. Increased temperature in the *s*-C_3_N_4_ nanoparticles with increasing laser intensity leads to the growth of the electrons in the first vibronic state (*n* = 1 in Figure 5) and, therefore, to an increase in anti-Stokes green luminescence [10]. Our hypothesis is in agreement with a recent finding that low-dimensional two-dimensional perovskite (C_6_H_5_C_2_H_4_NH_3_)_2_PbCl_4_ quantum wells are efficient in spontaneous anti-Stokes visual luminescence under visual or near-infrared laser excitation [26]. They demonstrate a luminescence upconversion process from the visible self-trapped exciton to an ultraviolet free exciton excited by the nanosecond pulse laser excitation. 

## 5. Conclusions

Fluorescence spectra of graphitic (*g*-C_3_N_4_) and spherical (*s*-C_3_N_4_) modifications of carbon nitride were measured as a function of green laser intensity. 

The intensity of the laser increases the maximum of the fluorescence shifts towards the anti-Stokes side of the fluorescence for carbon nitride hollow spheres. Such a phenomenon was not observed for graphitic carbon nitride particles. 

The maximum of the anti-Stokes fluorescence in *s*-C_3_N_4_ nanoparticles was observed at 480 nm. The anti-Stokes component had a minimum threshold of ~100 mW for this experimental setup. The Stokes (regular) component increases laser intensity up to 98 mW and then starts decreasing. The ratio of the intensity of the anti-Stokes peak (centered at 480 nm) to that of Stokes peak (centered at 582 nm) was measured to be I_484/582_ = 6.4 × 10^−3^ at a low level of intensity (5 mW) using a green pulsed laser, whereas it rose to I_484/582_ = 2.27 with a high level of laser intensity (1500 mW). 

Studies of the behavior of the fluorescence of the *s*-C_3_N_4_ nanoparticles excited by CW lasers with different wave lengths revealed that: (a) there is only one center of the emission peak at 480 nm when illumination of the *s*-C_3_N_4_ was conducted with 266 nm laser; (b) using a green laser (514 nm), the position of the peak of the fluorescence was around 582 nm. 

The slope of the log–log curve (0.96) for the intensity of the anti-Stokes peak leads to the conclusion that the thermal UC mechanism is responsible for the enhancement of the anti-Stokes fluorescence in *s*-C_3_N_4_ nanoparticles excited by a high intensity green laser.

## Figures and Tables

**Figure 1 nanomaterials-11-02529-f001:**
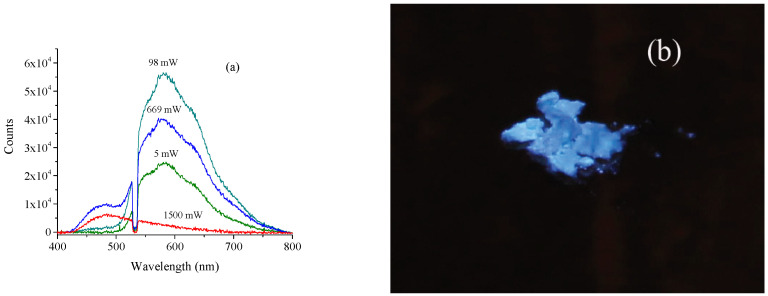
(**a**) Power dependence of the visible fluorescence signal of *s*-C_3_N_4_. Measured powers in mW are displayed in the table. (**b**) Photos of the *s*-C_3_N_4_ powder illuminated by a green laser 1500 mW.

**Figure 2 nanomaterials-11-02529-f002:**
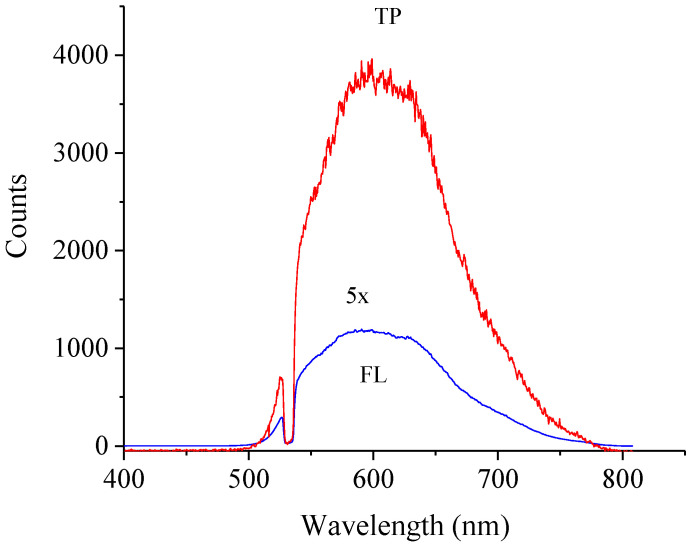
Fluorescence spectra of the *g*-C_3_N_4_ powder with 532 nm excitation, Nd-YAG laser, 15 Hz, 6 ns-pulse. The lower intensity spectrum was measured over one laser pulse and 5mW laser power. It was also scaled for camera gain and further scaled 5X for clarity. The higher intensity spectrum was measured over one laser pulse and 1.4 W laser power.

**Figure 3 nanomaterials-11-02529-f003:**
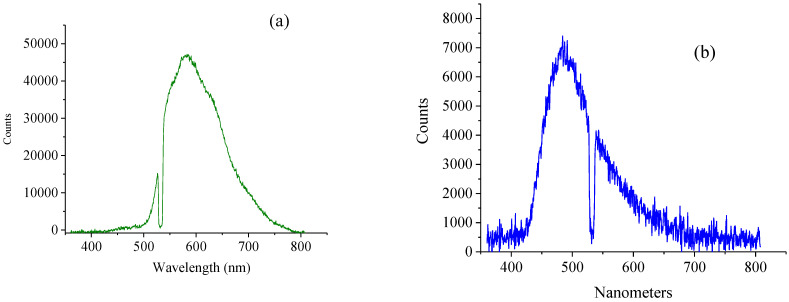
Spectra of the fluorescence of the spherical modifications by (**a**) green laser (514 nm) and (**b**) ultraviolet laser (266 nm).

**Figure 4 nanomaterials-11-02529-f004:**
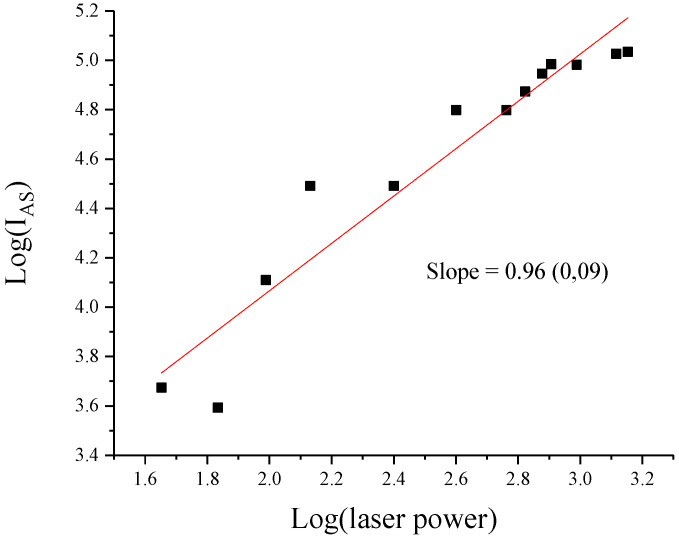
Linear log–log relationship between the anti-Stokes emission intensity and incident irradiance of the *s*-C_3_N_4_ nanoparticles. The number in parentheses is the standard deviation.

**Figure 5 nanomaterials-11-02529-f005:**
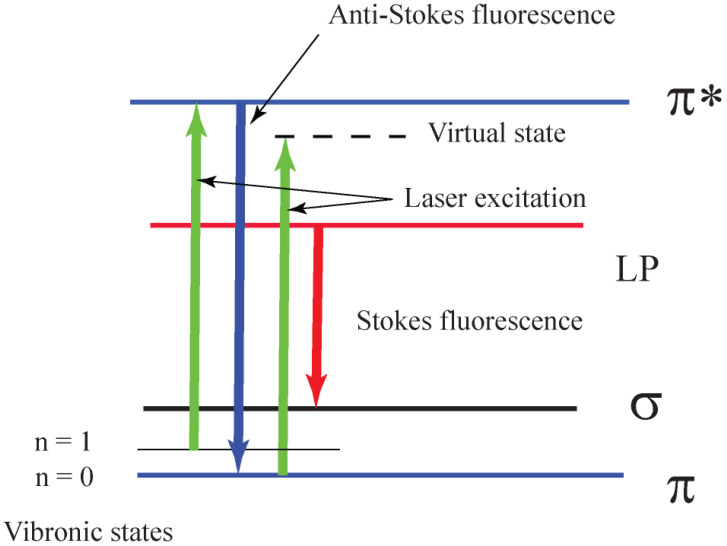
Schematic energy level diagram of the fluorescence emission from *s*-C_3_N_4._

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
