# Peer review of "Enhancement of the Anti-Stokes Fluorescence of Hollow Spherical Carbon Nitride Nanostructures by High Intensity Green Laser"

_nanomaterials, 2021, doi:10.3390/nano11102529_

Round 1

Reviewer 1 Report

Dear Editor

Journal of Nanomaterials

Manuscript " Enhancement of the anti-Stokes fluorescence of the hollow spherical carbon nitride nanostructures by high intensity green laser” by Pavel V. Zinin, Tayro Acosta, Anupam Misra and Shiv K Sharma, for Journal of Nanomaterials, investigates the non-linear effect on anti-Stokes fluorescence using a nanosecond pulsed laser excitation, at 532 nm, on different graphitic C3N4 structures including s-C3N4 structures and disordered 36 graphitic C3N4 (g-C3N4).

The paper sounds interesting for the possibility to exploit anti-Stokes fluorescence on carbon nitride samples. Text should be modified for typing errors and uncorrected expressions, and many points should be improved, to make the paper clearer and more understandable.

The abstract contains a typo in the instructions that should be deleted: “single paragraph of about 200 words maximum. “

The introduction and the conclusions must be carefully reviewed: they contain very fragmentary information and should be improved, should be described in more detail. Introduction presents some extra space and writing errors “… graphituc”.

Materials and Methods, lines 44-46, report: “According to the transmission electron microscopy data …”, the cited TEM data have been cited but not added to the manuscript, it should be interesting to observe directly the average diameter of samples, by reporting TEM images.

Materials and Methods, lines 47-48, reports twice the used wavelength of the laser source.

Caption of figure 1 reports “Measured powers in mW are displayed in the table”, but there is no table in the paper …

Materials and Methods, line 73 reports twice “the optical properties of the s-C3N4 and s-C3N4”; line 78 reports “wave lengths”; line 83 reports “valance”: all these points should be corrected

Caption of figure 2 is not clear, should be clarified. Authors affirm that a laser, characterized by a time duration of 6 ns and a repetition rate of 15 Hz, is used for the measurements. The measurements reported in figure are made using different gates of 200 ns and 500 ns, but using a pulsed laser with a frequency of 15 Hz, the choice of the different gate does not affect the number of pulses on the samples. So, it is not clear to the reader why were different gates chosen for the two measurements?

A scheme of the electronic levels of carbon nitride samples should be added in order to make the discussion of data clearer and more understandable.

Figure 4 reports “Linear log-log relationship between the anti-Stokes…”, in order to evidence tha quality of data and finning, error bar should be added to data and the estimation of the quality of the fitting should be defined

Finally, in the discussion part, a comparison with two-photon fluorescence measurements has been performed (line 87). It is not clear how the reported data can be compared with the two-photon fluorescence ones.

Author Response

We wish to express our gratitude to the Reviewers for the careful reading of the manuscript, the valuable comments and constructive suggestions provided. We have implemented the reviewer’s advice as follows:

Comment: The abstract contains a typo in the instructions that should be deleted: “single paragraph of about 200 words maximum. “

Reply: Corrected

Comment: The introduction and the conclusions must be carefully reviewed: they contain very fragmentary information and should be improved, should be described in more detail. Introduction presents some extra space and writing errors “… graphituc”.

Reply: The introduction has been reviewed and typos has been corrected

Comment: Materials and Methods, lines 44-46, report: “According to the transmission electron microscopy data …”, the cited TEM data have been cited but not added to the manuscript, it should be interesting to observe directly the average diameter of samples, by reporting TEM images.

Reply: In this report we used specimens provided us with Prof. Khabashesku. He produced then at the Rice University using the method carefully described in his publications. These publications contain a detailed results of the characterization of the particles by SEM/EDAX, TEM, XRD, EELS and FTIR spectroscopy. Therefore we did not conduct TEM characterization of Khabashesku’s specimens. However, in our previous study of the fluorescence of the s-C3N4 specimen we characterize the aggregation of the nanoparticles by the dynamic light scattering. This part of the Materials and Methods section was rewritten.

Comment: Materials and Methods, lines 47-48, reports twice the used wavelength of the laser source.

Reply: Corrected.

Comment: Caption of figure 1 reports “Measured powers in mW are displayed in the table”, but there is no table in the paper …

Reply: The sentence was deleted

Comment: Materials and Methods, line 73 reports twice “the optical properties of the s-C3N4 and s-C3N4”; line 78 reports “wave lengths”; line 83 reports “valance”: all these points should be corrected

Reply: Corrected.

Comment: Caption of figure 2 is not clear, should be clarified. Authors affirm that a laser, characterized by a time duration of 6 ns and a repetition rate of 15 Hz, is used for the measurements. The measurements reported in figure are made using different gates of 200 ns and 500 ns, but using a pulsed laser with a frequency of 15 Hz, the choice of the different gate does not affect the number of pulses on the samples. So, it is not clear to the reader why were different gates chosen for the two measurements?

Reply: We agree with reviewer. We use the gate width of either 200 ns or 500 ns in our experiments. The fluorescence has fast life time, so opening the gate to 500 ns does not make any difference in the signal intensity. Therefore, to avoid confusion we remove gate width from the caption, but leave them only in the Materials and Method section.

Comment: A scheme of the electronic levels of carbon nitride samples should be added in order to make the discussion of data clearer and more understandable.

Reply: A schematic energy level diagram of the fluorescence emission from s-C3N4was added to the manuscript (Fig. 5).

Comment: Figure 4 reports “Linear log-log relationship between the anti-Stokes…”, in order to evidence that quality of data and finning, error bar should be added to data and the estimation of the quality of the fitting should be defined

Reply: We did not conduct several measurements at each point on the log-log curve. However, it is not necessary to estimate the error of the slope of the line as far as number of measurements in higher than two. It is more advantages to make a single measurement at each point on the line in a longer range of laser power to get a good estimation of the statistical error of the slope of the curve. We agree that the value of the standard deviation of the slope on the graph in Fig. 4 was provided in the manuscript. We corrected this in the revised version (see Fig.4)    

Comment: Finally, in the discussion part, a comparison with two-photon fluorescence measurements has been performed (line 87). It is not clear how the reported data can be compared with the two-photon fluorescence ones.

Reply: We slightly corrected the discussion part to have a clear comparison. The comparison with the two-photon fluorescence was presented in the discussion. The slope of the log-log line of two photon fluorescence should be around two and that we mentioned in the discussion. We also provided corresponding references. 

Reviewer 2 Report

This paper report the anti-Stokes fluorescence of spherical (s-C3N4) carbon nitride increase with the intensity of the laser. Although the fluorescence of carbon nitride has been widely researched, the relationship of anti-Stokes fluorescence properties with the intensity of laser is unknown. Besides, it lacks the careful examination of the structure characterizations. Major revisions should be made before further consideration for publication in Nanomaterials.

  1. The structure characterizations of the as-prepared s-C3N4 are absence, such as TEM, XRD. It is not clear the difference between g-C3N4 and s-C3N4.
  2. Spelling mistakes should be avoided, for instance, page 1, line 30 “yas”, line 31 “graphituc”, line 33, nanoscale was repeated.

Author Response

We wish to express our gratitude to the Reviewers for the careful reading of the manuscript, the valuable comments and constructive suggestions provided. We have implemented the reviewer’s advice as follows:

Comment: The structure characterizations of the as-prepared s-C3N4 are absence, such as TEM, XRD. It is not clear the difference between g-C3N4 and s-C3N4.

Reply: In this report we used specimens provided us with Prof. Khabashesku. He produced then in Rice University using the method well described in his publications. These publications contain a detailed results of the characterization of the particles by SEM/EDAX, TEM, XRD, EELS and FTIR spectroscopy. Therefore we did not conduct TEM characterization of Khabashesku’s specimens. However, in our previous study of the fluorescence of the s-C3N4 specimen we characterize the aggregation of the nanoparticles by the dynamic light scattering. This part of the Materials and Methods section was rewritten.

Comment: Spelling mistakes should be avoided, for instance, page 1, line 30 “yas”, line 31 “graphituc”, line 33, nanoscale was repeated.

Reply: Corrected

Round 2

Reviewer 1 Report

The suggested modifications have been performed.

Reviewer 2 Report

no further comments